# Phase Behavior of Polydisperse Y-Shaped Polymer Brushes under Good Solvent Conditions

**DOI:** 10.3390/polym16050721

**Published:** 2024-03-06

**Authors:** Petr Fridrich, Zbyšek Posel

**Affiliations:** Department of Informatics, Faculty of Science, Jan Evangelista Purkyně University in Ústí nad Labem, 400 96 Ústí nad Labem, Czech Republic; petr.fridrich.1998@gmail.com

**Keywords:** Y-shaped brush, self-assembly, dissipative particle dynamics, ripple phase, surface

## Abstract

Y-shaped polymer brushes represent a special class of binary mixed polymer brushes, in which a combination of different homopolymers leads to unique phase behavior. While most theoretical and simulation studies use monodisperse models, experimental systems are always polydisperse. This discrepancy hampers linking theoretical and experimental results. In this theoretical study, we employed dissipative particle dynamics to study the influence of polydispersity on the phase behavior of Y-shaped brushes grafted to flat surfaces under good solvent conditions. Polydispersity was kept within experimentally achievable values and was modeled via Schulz–Zimm distribution. In total, 10 systems were considered, thus covering the phase behavior of monodisperse, partially polydisperse and fully polydisperse systems. Using such generic representation of real polymers, we observed a rippled structure and aggregates in monodisperse systems. In addition, polydisperse brushes formed a stable perforated layer not observed previously in monodisperse studies, and influenced the stability of the remaining phases. Although the perforated layer was experimentally observed under good solvent conditions and in the melt state, further confirmation of its presence in systems under good solvent conditions required mapping real polymers onto mesoscale models that reflected, for example, different polymer rigidity, and excluded volume effects or direct influence of the surface, just to mention a few parameters. Finally, in this work, we show that mesoscale modeling successfully describes polydisperse models, which opens the way for rapid exploring of complex systems such as polydisperse Y-shaped brushes in selective or bad solvents or under non-equilibrium conditions.

## 1. Introduction

Polymer brushes are polymer chains tethered to a surface or an interface whose behavior, e.g., assembly into a nanostructure, modifies their surface properties. Among many different variations of brushes, multicomponent systems contain two or more polymer chains grafted onto the surface [1]. Due to the chemical diversity of chains, their lengths, grafting densities, responsiveness to the environment, and polydispersity, etc., the chains organize or assemble onto the surface, thus creating rich phase behavior. This phenomenon has many technological applications including in manufacturing smart/responsive surfaces that have tunable mechanical, optical or electrical properties [2,3].

Binary mixed polymer brushes represent two chemically different homopolymers grafted to the surface by one end. In these systems, the grafting density, mutual chain lengths and incompatibility are the usual parameters used for controlling the structure assembled on the surface [4]. A special class of these brushes is the Y-shaped brushes, in which each pair of incompatible polymers is grafted to the same point by using specified precursors [5,6]. Among other uses, such a grafting strategy is useful for achieving more uniform distribution of different polymers on the surface and prevents unwanted fluctuations in density of each specie. In Y-shaped brushes, a large variety of surface structures has been described so far, including structures that are rippled or dimpled, aggregates, and pinned micelles, etc. The presence and stability of the structure depends mostly on typical parameters such as mutual chain length, the uniformity of grafting points distribution, solvent quality and selectivity or chemical diversity of attached polymers.

Within theoretical and simulation studies, Zhulina et al. [7] developed a theoretical model using scaling arguments to describe the phase behavior of A/B incompatible Y-shaped brushes with the same chain lengths. By changing the grafting density from low to high, the model predicts transition between different structures in melt and under nonselective poor solvent conditions. The model predicts the presence of pinned micelles and brushes. Later, Minko et al. [8] combined SCF calculations to construct a phase diagram of mixed polymer brushes that are grafted randomly onto the surface with the same grafting density and have an identical chain length. Beside disorder configurations, they observed, under good solvent conditions, a rippled structure, while selective conditions produced a dimpled structure. Later, they compared their results with those of PSF and PMMA brushes exposed to different solvents and observed similar behavior as that predicted by SCF. Then, Yin et al. [9] used a bond fluctuating model and annealing technique to describe the lateral and perpendicular segregation of Y-shaped brushes composed of ABC triblock copolymer, whose middle part was defined by a short block that was grafted onto the substrate. The lengths of A and B polymers were equal. The phase behavior in nonselective (poor and good) and selective (to A) solvents has been reported, describing segregation of brushes to the zoo of configurations such as mixed or core-shell micelles, internally segregated micelles, hamburger-like micelles, segmented wormlike micelles, connected micelles and split micelles. Due to low grafting density, the rippled structure has not been reported. Wang et al. [10] applied coarse grained modeling and an implicit solvent model to describe microphase separation in mixed brushes exposed to various solvents. They systematically investigated the dependence of the morphology of mixed polymer brushes on a series of parameters including brush composition, individual grafting densities, solvent selectivity, etc. Proper identification of different morphologies that lacked long-range ordering was achieved by Minkowski measures. In their symmetric mixed brushes, the rippled structure was observed in systems under good solvent conditions. The effect of polydispersity on conformation of spherical brushes was reported by Dodd et al. [11] using Monte Carlo simulations as a function of grafting density and the polydispersity index (PDI). The authors considered low and intermediate grafting densities and PDIs in a range from low to high values (2.5). The simulation results suggested that at high PDI, the chains, which were shorter than the average value, were more compressed at the surface, thus filling the voids accessible to free polymer ends, and that longer chains increased the thickness of the grafted layer. Moreover, additional authors devoted their efforts to modeling the effect of polydispersity on the assembly of copolymers. Among them, Jiao et al. [12] showed that polydispersity of diblock copolymers can stabilize bicontinuous structures in a wide range of chain compositions of polydisperse chains, and that phase diagrams produced by a combination of monodisperse and polydisperse chains lose their symmetry around $f_{B}=0.5$ and shift order–disorder transition to higher values. Then, Qi et al. [13] used Monte Carlo simulation, field theory and analytical theory to study the effect of polydispersity on the internal structure of polymer brushes. Their simulation revealed that increasing the PDI changed the shape of the brush density profile from convex to concave and that the free chain end fluctuations were significantly suppressed even at low PDI. In addition, Klushin et al. [14] studied the structure of a polydisperse brush at low PDI with linear brush density profiles. In most cases, the polydispersity was modeled by Schulz–Zimm distribution [15,16]. Later, Gao et al. [17] again used a monodisperse model and dissipative particle dynamics to study formation of a thin brush layer of Y-shaped brushes under different solvent conditions at different grafting densities and at different chain incompatibilities. At low grafting densities, e.g., in the brush mushroom regime, they observed structures similar to those in [9]. Stripe-like structures, e.g., ripple phase, was observed at higher grafting densities in brush regime. More importantly, Yin et al. [18] studied the influence of grafting point distribution on the formation and ordering of Y-shaped brushes on flat surface using same technique as in [9]. They showed that both randomness and distance distribution of grafting points influence not only the dimensions of ripple structure but also controls its long-range ordering. Recently, Miliou et al. [19] used molecular dynamics to describe the influence of salt free, monovalent or divalent solution of oppositely charged Y-shaped brushes. Their simulation showed that lamellar structure is formed by symmetric brushes dissolved in divalent salt. For asymmetric brushes at high salt concentrations the isolated cylinders (aggregates) are formed.

Recently, we used dissipative particle dynamics to describe the self-assembly of Y-shaped brushes grafted onto flat surface [20]. In our modeling, we considered homopolymer brushes and brushes with low polydispersity (not higher than 1.1) and vary the grafting density and mutual chain length of homopolymers. Results showed that mesoscale model was able to follow proper scaling behavior of brush height and that typical structures like aggregates and ripple phase are observed under good solvent conditions. In addition, we reported, for the first time, formation of perforated layer in low polydisperse brushes.

Within the experimental framework, Julthongpiput et al. [21] designed Y-shaped brushes composed of hydrophobic and hydrophilic chains represented by PSA and PBA and observed pinned micelles. Moreover, by changing the environment from hydrophilic to hydrophobic, they observed local reversible rearrangement in different solvents. Later, Zhang et al. [22] synthetized Epoxy-Based Block of Poly (glycidyl methacrylate) Y-shaped brush grafted onto wafer surface and characterized both thermal and structural properties of the brush layer. Moreover, self-assembly of the brushes were observed at different solvent conditions and morphologies like spheres, worm-like patterns, nanowell patterns, and dendritic patterns were obtained that could match aggregates, ripple structure as well as perforated layer discussed later in our Results section. The polydispersity was kept at lower (1.27) and moderate values (1.5). Bao and coworkers [23] systematically studied the effect of grafting density on microphase separation of silica particles grafted by poly (tert-butyl acrylate)/polystyrene Y-shaped brushes. The study proved that overall grafting density of the brush can be tuned keeping individual grafting densities the same. Overall grafting density decreased from 1.06 to 0.122 chains/nm^2^ and the polydispersity of both homopolymers was kept low, smaller than 1.11 for PtBA and smaller than 1.25 for PS. Results showed that under good solvent conditions the ripple structure formed above 0.34 chain/nm^2^ grafting density and was stable until 1.06 chains/nm^2^. Contrary, while decreasing the grafting density to 0.122 the phase separation disappeared. Later, the grafted silica particles were placed also to nonselective poor solvent environment [24] where the brushes formed nearly bicontinuous rippled nanopatterns whose size increased with increasing the brush chain length. Tonhauser et al. [25] used junction-point-reactive block copolymers to prepare surfaces with reversibly switchable wettability by using PS-PEO Y-shaped brushes. The modified silica surface was placed into different solvents to produce stimuli-responsive behavior of hydrophilic and hydrophobic chains. These either collapse onto the surface or dissolve in the solvent. The reversible behavior was measured mainly by contact angel measurements. The grafting density was kept low (no higher than 0.223 chains/nm^2^) and polydispersity did not exceed 1.17. Later, Huang et al. [26] fabricated Y-shaped brush of PEG and fluorinated PMMA with short attachment to the surface and studied its protein-resistance performance. They showed that while PEG chains dissolve in the environment, the fluorinated PMMA domain remains relatively stable thus preventing the adsorption of proteins on modified surface. The polydispersity was kept lower that 1.21 and AFM images suggest that mostly aggregates and ripple structures were observed. The application of Y-shaped brush for pH controllable oil/water separation was studied by Li et al. [27] The polydimethylsiloxane-block-poly(2-hydroxyethylmethacrylate)-block-poly(2-(dimethylamino)ethyl methacrylate) triblock copolymer was grafted onto the surface. Such functionalized surface was able to separate efficiently different water/oil mixtures by combining the pH responsiveness of PDMAEMA and hydrophobicity and flexibility of PDMS. Wei et al. [28] studied assembly behavior of different PMMA/PS mixed brushes at high grafting density (0.9 chains/nm^2^) on planar substrate. By modulating the length of PS chain, while keeping the grafting densities equal, they observed transition from disorder state to various structures. Phase diagram and SEM images suggest that aggregates (cylinders), ripple structure and perforated layer phase discussed later in our text are observed. This suggestion is further supported by the fact that phase diagram is not symmetric which can be also attributed to different PDI of PMMA and PS. Recently, Liu et al. [29] synthetized silica particles grafted by Y-shaped brushes with coassembly approach. The PEG/PS brush has been used to assembly the surface structures (aggregates, ripple or layer) that controls spatial organization of silica particles. Targeting the surface structure can contribute to controllable fabrication of hierarchical nanostructures on solid surfaces.

Reviewing the literature showed that while experimental Y-shaped brushes always possess certain polydispersity, in range between 1.05–2.5, the theoretical and simulation papers mostly work with monodisperse systems. Moreover, experimental systems prefer more symmetric polymers in the brush while theoretical models can easily change the lengths of individual polymers including its grafting density. Therefore, linking between theoretical and experimental results, in terms of structures, is sometimes hard.

Previously, we showed that mesoscale model is able to describe correctly the behavior of monodisperse Y-shaped brushes under good solvent conditions in terms of structures and scaling behavior. Moreover, we reported formation of perforated layer for brushes with low polydispersity, e.g., 1.1, at different grafting densities [20]. This structure has not been observed before in monodisperse systems. In this work, we fixed the grafting density at σ=0.5 chains/area and extend our modelling to systems with intermediate and highly polydisperse brushes. The values of polydispersity are kept in experimental range where PDI∈<1.1, 2.0>. To obtain full phase diagrams we also change the polymer chain architecture, e.g., $f_{A}$, from 0.1 to 0.9. In overall, 10 systems are considered in this study covering monodisperse, partially polydisperse and fully polydisperse systems.

First, we describe our simulation model used previously and present the distribution of chain lengths, e.g., PDI values, in our model. Then, we explain our detailed analysis of assembled structures and workflow for phase diagram construction. In the Results section, we first refresh previously modeled monodisperse system at relevant grafting density. Then, we show and discuss results for different combinations of homopolymer’s polydispersity values. We first discuss partially polydisperse systems where one homopolymer is kept monodisperse while the remaining one possess polydispersity greater than 1.0. Here, the presence and stability of perforated layer is discussed in detail. Then, both homopolymers can be polydisperse and corresponding phase diagrams are shown. Finally, the influence of polydispersity on assembly of chains and stability of individual phases is discussed and concluded. Additional results referring to identification of equilibrium structure, distribution of chain lengths and configuration snapshots are shown in Appendix A.

## 2. Materials and Methods

In this section, we first describe Y-brush mesoscale model and inclusion of polydispersity into the monodisperse system. Then we focus on modelling technique and present the necessary simulation details. Finally, we present main observables that are used beside visual inspection of configuration, to identify and distinguish the equilibrium structures. Additional details like list of abbreviations used further in the text, modeling, parameters, proper identification of assembled structure are reported in Appendix A.

### 2.1. Mesoscale Model of Polydisperse Y-Shaped Brush

Figure 1 shows schematic picture of three different Y-shaped brushes grafted onto flat surface. Y-shaped brush is composed of two different homopolymers, A and B, that are attached to same grafting point. The variation in homopolymer chain length ratio is described by polymer chain architecture $f_{A}=N_{A}/\left(N_{A}+N_{B}\right)$ where $N_{A}$ and $N_{B}$ are A and B homopolymer chain lengths, respectively. In monodisperse systems, the $N_{A}$ and $N_{B}$ represent exact chain lengths of polymers. In polydisperse systems, the values represent average chain lengths. For additional details see Equation (1) and related discussion, Appendix A. Each homopolymer in the brush is represented by collection of beads connected by harmonic springs. Then each pair of A/B homopolymers is grafted to the same grafting point. Grafting points are distributed randomly on the surface. The surface itself is represented by collection of beads that do not move throughout the simulation, e.g., they are “frozen”. To avoid unphysical fluctuations of the density near the solid wall, the density of the wall beads matches the density of the polymer solution. Moreover, to avoid penetration of solvent or brush beads into the surface, the reflective layer with bounce-back boundary conditions [30] is placed on the interface between wall and fluid. Solvent is represented by single beads (not displayed here).

The well-known Schulz-Zimm distribution [15,16] is used to include polydispersity into the monodisperse brush. This distribution describes reasonably the distribution of polydisperse chains in the experiment and can be calculated by Equation (1)
(1)$$P\left(N\right)=\frac{k^{k}N^{k-1}}{\Gamma (k)N_{n}}e^{\left(-k\frac{N}{N_{n}}\right)}$$
where $\Gamma \left(k\right)$ is the delta function, $N_{n}$ is the number averaged chain length, $N_{w}$ is the weight average chain length and k controls the polydispersity index PDI via relation $PDI=N_{w}/N_{n}=1+1/k$. For monodisperse systems, the k → ∞, while for highly polydispers systems the k→1. In our modelling, we limited our polydisperse chain lengths to be in range N ∈<5,180> beads in one chain, to meet reasonable computational criteria. Representative chain length distributions used in our modelling are shown in Appendix A. Moreover, we present simplified workflow for generating the distribution of A/B chains with predefined values of polydispersity in Appendix A.

### 2.2. Mesoscale Modeling and Simulation Details

Well-established Dissipative Particle Dynamics (DPD) [31] is used to generate simulation trajectories in this study. DPD has been previously used to describe systems that self-assemble at nanoscale within bulk block copolymers [32,33], copolymers grafted to flat and curved surfaces [34,35] or copolymers with well-defined monomer sequence [36] and under non-equilibrium conditions [37], just to mention few works related to modelling of Y-shaped brushes.

In our DPD, we adopt standard reduced units with $k_{B}T$ being the unit of energy, where $k_{B}$ is the Boltzmann constant and T is the thermodynamic temperature. Cutoff distance $r_{c}$ and mass m of a bead then represent unit of length and mass, respectively. All beads in our model poses same mass and volume. The total reduced bead density is set to $\rho r_{c}^{3}=3.0$. All beads interact with standard DPD potential [38], where $\left(a_{AB}r_{c}\right)/\left(k_{B}T\right)$ is the maximum repulsion between unlike beads and is related to the standard Flory-Huggins interaction parameter $\chi_{AB}$. Additional details about DPD are shown in SI. In addition, polymer beads are connected by harmonic spring described by force in Equation (2)
(2)$$f_{i,i+1}^{bond}=K_{s}\left(r_{i,i+1}-r_{0}\right)$$
where $K_{s}=4k_{B}T$ is the stiffness of the spring, $r_{i,i+1}=\left|r_{i+1}-r_{i}\right|$ and $r_{0}=0\;r_{c}$ is the equilibrium distance of the spring. Stronger spring force ($K_{s}=100k_{B}T$ and $r_{0}=0.5\;r_{c}$) also binds homopolymers to the surface.

All simulations are performed in LAMMPS simulation package [39] with GPU acceleration [40]. All initial structures are prepared by in-house developed codes in Python Fortran and MATLAB languages. Moreover, all calculations of observables are done by in-house developed code in Python. For all system, we follow same simulation protocol where all simulations start from random initial configuration with all interactions set to $\left(a_{ij}r_{c}\right)/\left(k_{B}T\right)=25$. After initial mixing, the interaction parameter between A and B is gradually raised with ${∆a}_{AB}=5$ up to final value 50. Remaining interaction parameters are kept at 25. Each raise of interaction parameter is followed by equilibration period followed by production run where we sample the trajectory. If the structure does not assemble until final interaction value, the system is marked as disordered. For each production run we collect 5000 uncorrelated configurations for calculating the observables. At least $10^{4}$ simulation steps separate each stored configuration and ensures that collected samples are uncorrelated.

For all systems, we use simulation box with two flat surfaces placed at the bottom and on the top of z-axis. The periodic boundary conditions are imposed only in x,y-plane. The flat surface has $L_{x}=L_{y}=60r_{c}$ and only the bottom one is grafted by brushes. The thickness of the surface is $3r_{c}$. The top surface servers only for keeping the solvent inside the simulation box. For monodisperse systems, we use cubic simulation box with $L=60r_{c}$. In polydisperse systems, the $L_{z}$ is kept big enough to avoid interaction of the longest chain with the top surface. We adopt (and tested) the simple rule $L_{z}=5R_{g}$, where $R_{g}$ is radius-of-gyration of the longest chain in the system. Including also the solvent and wall beads, we model systems whose size spans from $6\cdot 10^{5}$ up to $4\cdot 10^{6}$ beads in total. Snapshots of two different simulation boxes are shown in Appendix A.

Unlike in our previous study [20], here we fix the grating density to $\sigma =0.5\;\frac{chains}{area}$ and consider following polydispersity ${PDI}_{A/B}=\left\{1.0,1.1,1.5,2.0\right\}$ of A/B homopolymers. Moreover, we systematically vary the A/B homopolymer chain length ratio, $f_{A}$, from 0.1 to 0.9 to obtain full phase behavior. In overall, we model 10 systems with different combinations and different $f_{A}$. We considered only those combinations of PDIs where ${PDI}_{A}\leq {PDI}_{B}$. Table 1 shows all different combinations of PDI considered here together with labels used for systems further in the text.

### 2.3. Observables

Simulation observables are mainly chosen to describe the assembled structures and are obtained on the scale of individual chains as well as on the scale of the structure itself. On the scale of individual chains, we calculate mainly the height of the brush H.
(3)$$H(z)=2\frac{\int_{0}^{L_{z}}\rho \left(z\right)z\;dz}{\int_{0}^{L_{z}}\rho \left(z\right)\;dz}$$
where $\rho \left(z\right)$ is the density profile perpendicular to the solid surface. This experimentally achievable feature is later used in Minkowski and DBSCAN calculations. Moreover, we evaluate also other standard chain properties including end-to-end distance and radius-of-gyration. Lateral structure factor ${SF}_{xy}\left(q\right)$, Equation (4), is calculated to describe the organization of chains on the surface
(4)$${SF}_{xy}\left(q\right)=\frac{1}{N}\left[{\left(\sum_{i}\cos\left(q\cdot r_{i}\right)\right)}^{2}+{\left(\sum_{i}\sin\left(q\cdot r_{i}\right)\right)}^{2}\right]$$
where q is the wave vector, $r_{i}$ is the position of i-th bead in the simulation box and N is the total number of A or B beads.

More importantly, we use Minkowski measure to distinguish individual structures. These functionals has been previously used for example for description of dynamics in block copolymer microdomains [41] or for studying the formation of structure in thin films [42]. Three Minkowski functionals (Surface area $M_{S}$, boundary length L and Euler characteristic E) are typically used for characterizing 2D objects that lack long-range ordering. They have already been successfully applied in monodisperse brush systems before and we follow the procedure described in [10]. Here, we use only the Minkowski measure $M_{S}$ representing the area covered by chains of type A. To calculate the $M_{S}$ properly, the 3D density distribution is transformed into 2D map that is then binarized to black and white image. In monodisperse and polydisperse systems, we transformed the spatial density into two dimensions following procedure described in [10]. Moreover, since our systems are dispersed in good solvent, we calculate the composition contrast $\rho^{-}=\rho_{A}-\rho_{B}$ for systems with same chain lengths. For other cases, the distribution depends on distance z from the surface and we use the composition contrast $\rho^{-}\left(z\right)$, where the distance from the surface is given by height of the brush H. To binarize the 2D compositional contrast map into black and white image we use the threshold ξ=0 for all cases. The choice of the threshold value ensures that Minkowski functionals still characterize the domain structures instead of density fluctuations inside the domains themselves [10]. Binary images are then adjusted by morphological operators to minimize the errors that stems from reducing the dimensionality. The Minkowski functionals including calculation of density maps, binarization and filtration were calculated by in-house developed code following the procedure described in reference [10].

Finally, we employ clustering technique DBSCAN [43] to distinguish transitions between different structures in phase diagram and to evaluate the solidity of polydisperse structures. DBSCAN construct clusters from points that are closely packed together in terms of many near neighbors and identify outliers that lies in low density regions. For clustering, we consider all segments of type A within the $L_{x}\times L_{y}\times 3r_{c}$ box. The origin of the box is located at the Brush height H where the structure is supposed to be least influenced by surface as well as by polymer fluid interface. The radius of clustering ε is maintained 0.9 for all system with monodisperse chains and 1.2 for systems with fully polydisperse chains. In both cases, we set 10 points to be the minimum number for cluster formation. The DBSCAN calculations were done using the scikit-learn Python package (v. 1.4) [44].

## 3. Results

In all systems described further, we keep ${PDI}_{A}\leq {PDI}_{B}$ (see Table 1). First, we present and discuss phase behavior of systems where A homopolymer is kept monodisperse and B homopolymer vary in PDI (S1–S4 systems). Then we let the A homopolymer be polydisperse and describe the systems that have different ratios of polydispersity (S5–S10). In all phase diagrams, we describe the phase behavior in $f_{A}-a_{AB}$ plane, where $a_{AB}$ can be matched with experimentally achievable $\chi_{AB}$ if needed. Moreover, we describe the conditions that are necessary for formation of perforated layer and discuss its stability with respect to system conditions like PDI and $f_{A}$. Additional results including Minkowski parameters and clustering behavior that are used for constructing the phase diagrams are shown in Appendix A.

### 3.1. Phase Behavior of Monodisperse and Partially Polydisperse Y-Shaped Brushes

Figure 2 shows phase diagrams for systems where A homopolymer is kept monodisperse and polydispersity of B homopolymer vary from 1.0 up to 2.0. Phase diagrams in Figure 2a,b were reprinted with permission from our previous study [20] and are shown here for comparison with remaining systems considered further in the study. Figure 2a show monodisperse system S1. Figure 2b–d show phase behavior of S2–S4 systems with partially polydisperse brushes. Figure 2a,b were reprinted with permission from our previous study [20]. Figure 2a shows typical phase behavior of monodisperse system where only ripple structure and aggregates are formed. This result agrees with previous theoretical and modeling studies [8,9,10,17,18,28]. In addition, Zhang et al. [22] studied Y-shaped Epoxy-Based Block of Poly (glycidyl methacrylate) Y-shaped brush grafted onto wafer surface under various solvent conditions. Under good solvent conditions (THF or DMF in their case), the increase in brush concentraion led to the transition from aggregates (isolated cylinders in their case) to ripple structure (please see Figures 18 and 19 in the reference). Later, similar structures has been observed by Wei et al. [28] who studied assembly of different PMMA/PS mixed brushes at high grafting density (0.9 chains/nm^2^) on planar substrate in melts state (Please see for example SEM images in Figure 3 or images in Figure 4 in the reference). Representative configurations from our model are for comparison shown in Appendix A.

We see that polydispersity of B homopolymer triggers the formation of perforated layer (PL). Indicative configurations can again be seen in Appendix A. The mechanism and conditions of its assembly are discussed in detail in following text. Different colors in polydisperse phase diagrams (Figure 2b–d) represent phases formed by A (red) or B (blue) homopolymers, i.e., minority component, that forms the structures. Both, A and B, contribute significantly to formation of PL, thus we label it as a red triangle filled with blue color.

In partially polydisperse systems (S2–S4), we see that aggregates are fully preserved on the edges of phase diagram only for least polydisperse systems S2 (Figure 2b). Increasing the ${PDI}_{B}$ to 1.5 neglect formation of aggregates by polydisperse chains (see $f_{A}=0.9$ in Figure 2c) and further increase to 2.0 expels aggregates completely (see $f_{A}=0.1$ and $f_{A}=0.9$ in Figure 2d). On monodisperse side of phase diagram ($f_{A}=0.1$), the reason stems from presence of short polydisperse chains that organize closer to the surface and forces short monodisperse chains to assemble to ripple phase due to the high repulsion between A and B. On polydisperse side of phase diagram ($f_{A}=0.9$) the long polydisperse chains can interconnect isolated aggregates and form ripple structure instead. Beside dominant ripple phase, PL phase emerges at $f_{A}=0.5$ in Figure 2b and propagates further to polydisperse part of phase diagram. Here, the A homopolymer is supposed to form a compact layer above the ripple phase of B type. Nevertheless, long polydisperse chains penetrate this compact layer and form perforations. Increasing the ${PDI}_{B}$ to 1.5 and 2.0 Figure 2c,d, leads to extension of PL phase window up to $f_{A}=0.7$. Finally, increasing the ${PDI}_{B}$ to 2.0 shifts the order-disorder transition to higher $a_{AB}(\chi_{AB})$ values.

Beside visual inspection and structure factor, we determine the morphology by Minkowski measure $M_{S}$ and by clustering behavior obtained from DBSCAN. Both are shown in Figure 3. Left part of the figure shows $M_{S}$ values for all above-mentioned systems (S1–S4). Dotted line with opened circles represents monodisperse system S1 and due to the symmetry, the results are displayed only up to $f_{A}=0.5$. Results for S2–S4 systems are shown in full range. For all systems, increasing the $f_{A}$ increase also value of $M_{S}$ and leads to the plateau around $f_{A}=0.8$ where the A homopolymer forms compact layer above shorter B chain, regardless the value of PDI. Combination of $M_{S}$ and visual inspection allowed us to draw approximate boundaries (dashed line) in phase diagrams and define values for different structures. Proper identification of individual structures is further supported by plotting the number of clusters within a part of the trajectory (see right side of Figure 3). The profiles are shown for $f_{A\;}=\{0.1,\;0.3,\;0.6,\;0.9\}$ for S3 system, where $f_{A}=0.1$ represent aggregates, $f_{A}=0.3$ ripple structure, $f_{A}=0.6$ perforated layer and $f_{A}=0.9$ compact layer, respectively. Appendix A shows configurational maps obtained in the distance of Brush height H that are partitioned by DBSCAN to clusters. Finally, we also plot Structure factor ${SF}_{xy}(q)$ profiles to gain an insight into lateral ordering of chains. Indicative profiles are shown in Appendix A. Such combination of characteristics allowed us to reliably identify assembled structures also in less ordered system formed by fully polydisperse brushes.

#### Perforated Layer

Before discussing fully polydisperse systems let us briefly describe the mechanism that triggers formation of PL phase. As mentioned earlier in the text, when the system reaches the $f_{A}=0.5$ the monodisperse system forms ripple structure. Nevertheless, in polydisperse systems, the compact layer is formed at $f_{A}\;\geq \;0.5$ by monodisperse part (A) and the polydisperse part (B) self-assemble closer to the surface. Higher repulsion between A and B drives longer polydisperse chains to either pack onto the surface or to migrate to the good solvent that is above the compact layer. While space close to the surface is highly occupied by short B chains the longer ones are forced to stretch, penetrate the A layer and migrate to the solvent. These longer chains create a space for additional B chains thus creating stable perforations in the layer. Representative configurational snapshot of perforated layer is shown in Figure 4b,c from two different views. We speculate that regularity of perforations is given by regularity of grafting and that for randomly grafted chains the regular distribution of perforations cannot be achieved as discussed before by Yin et al. [18] for ripple structure. According to that, we plot in Appendix A different configurations of partially polydisperse system S2 (${PDI}_{A}=1.0,\;{PDI}_{B}=1.1$) together with brush grafting points. To better visualize the relationship between position of grafting points and assembled structure we plot only $30r_{c}\times 30r_{c}$ sample of the surface. Full configurations are shown in Appendix A.

Similar perforated layers as described here were experimentally observed again by Zhang et al. [22] by dissolving the Epoxy-Based Block of Poly (glycidyl methacrylate) Y-shaped brushes in DMF solvent (Please see Figure 11 in the reference). Moreover, similar structure has been also reported by Wei et al. [28] for PMMA/PS mixed brushes in the melt state (Please see Figure 3 in the reference). Therefore, we can speculate that formation of PL phase is mainly driven by interchain interactions then by other parameters. Nevertheless, to confirm this hypothesis more precise modeling, that spans the scope of this study, that maps real polymers onto the models is needed.

### 3.2. Phase Behavior of Fully Polydisperse Y-Shaped Brushes

Here, we investigate the influence of increasing polydispersity of A homopolymer on stability of individual phases observed either in fully monodisperse (S1) or in partially polydisperse systems (S2–S4). For further comparison of systems, we first discuss the solidity of structures assembled by fully polydisperse chains on the surface (S5–S10) by comparing mean number of clusters obtained from DBSCAN analysis. Then we present only those phase diagrams whose number of clusters does not deviate much from monodisperse case. Other systems are rejected from further discussion since their structures are highly disturbed and cannot be reliably categorized by observables considered here. Mean number of clusters for all systems is shown in Appendix A. We see that all fully polydisperse systems that have ${PDI}_{A}\;\geq \;1.5$ (S8–S10) deviates significantly from other systems. Bottom part of the Figure shows corresponding configurations. High polydispersity of these systems prevents detailed categorization of structures into groups and therefore these systems were rejected from further analysis. Therefore, Figure 5 shows phase behavior of systems S5–S7, where ${PDI}_{A}$ is fixed at 1.10 and ${PDI}_{B}$ changes from 1.10 up to 2.0. Comparison of these phase diagrams with monodisperse system S1 in Figure 2, shows that increasing the PDI changes the phase behavior in a significant way. Assembly of low polydisperse systems with ${PDI}_{A}={PDI}_{B}=1.10$ (S5) is shown in Figure 5a. Comparison with S1 system reveals that this system is much closer to monodisperse case than to partially polydisperse systems and that PL phase is not observed. Moreover, the same polydispersity of both homopolymers prevents formation of aggregates in both phases (A/B). Interestingly, increasing the difference between PDIs in S6 and S7, Figure 5b,c brings PL back to the game and the phase behavior is similar to partially polydisperse systems S3 and S4 in Figure 2c,d. Nevertheless, fully polydisperse chains suppress here formation of ripple structure and prefers aggregates in S6. Increasing the ${PDI}_{B}$ up to 2.0 in S7 promotes formation of aggregates even more but does not contribute to wider PL phase space window as is observed in S4 system.

Comparison between monodisperse (S1), partially polydisperse (S2–S4) and fully polydisperse (S5–S7) brushes shows that different combinations of ${PDI}_{A/B}$ leads to different assemblies and influence borders of different structures or even their presence. Results shown that PL phase is formed only in systems where ${PDI}_{A}\neq {PDI}_{B}$. Wider phase window of PL is observed in system with combination of monodisperse and polydisperse polymers. In systems, where both polymers are polydisperse, the PL phase window does not change with different PDI values.

Moreover, the formation of phases on both ends of phase diagrams is highly influenced by differences in PDIs. While in partially polydisperse systems the aggregates are replaced by ripple structure the opposite phase behavior is observed for fully polydisperse brushes. The reason stems from highly overlapping chain size distribution given by PDI. In monodisperse or partially polydisperse systems, the long and short chains of different types are well separated thus forming their regular phases or are forced to form more compact phases like ripple one. In fully polydisperse systems, both types of homopolymers contribute to long and short chains thus influencing formation of structure close to the surface as well as on the brush/solvent interface. Presence of short chains in compact layer together with high incompatibility prevents formation of larger structures and prefers isolated aggregates instead. Same reason hampers formation of stable structures when both PDIs are above 1.5. At that point the structure highly fluctuates in size and shape and is impossible to determine whether it forms aggregates or ripple phase.

## 4. Conclusions

In this work, we employed Dissipative particle dynamics to investigate the influence of polydispersity on assembly of Y-shaped polymer brushes on flat surfaces under good solvent conditions. The polydispersity was modelled by Schulz-Zimm distribution where we considered systems with polydispersity spanning from low (1.1) to high (2.0) values. Beside PDI we varied also individual homopolymer lengths to obtain full phase behavior. We considered monodisperse, partially polydisperse and fully polydisperse systems. In partially polydisperse systems, the A homopolymer was kept monodisperse while B homopolymer varied in PDI. In fully polydisperse systems both polymers possess polydispersity. To distinguish different configurations, we employed a group of observables. Beside structure factor, we used Minkowski and DBSCAN to reliably categorize also less ordered systems.

Contrary to our previous study, here we fixed the grafting density at σ=0.5 chains/area and systematically varied the polydispersity of homopolymers in the brush. First, we refreshed the previous results of monodisperse and partially polydisperse systems with low polydispersity, e.g., PDI≤1.1. Monodisperse systems correctly formed aggregates and ripple structure predicted by theory and also observed in previous simulation studies. In low polydisperse systems, we previously showed formation of perforated layer. By increasing the PDI here, we showed increased stability of perforated layer, widening of its phase space window and narrowing of ripple phase space window. The role of polydisperse chains in this replacement was discussed as well as linking the presence of perforated layer with previous experimental studies.

Finally, we modelled systems where both homopolymers were polydisperse. Comparison of these systems with monodisperse or partially polydisperse counterparts shown that perforated layer is not formed when polymers poses same polydispersity. On the other hands, difference in PDI led to stable PL phase window that was slightly wider in partially polydisperse systems than in fully one. Moreover, full polydisperse systems prefer formation of aggregates on the right side of phase diagram instead of ripple phase and this trend is more pronounced for higher differences in PDIs. We saw that increasing both PDIs above 1.5 led to high fluctuations in structure, especially close to the polymer/fluid interface and hampered the identification of stable structure.

Overall, we successfully modelled influence of polydispersity on formation and stability of individual phases of Y-shaped polymer brushes grafted on flat surface under good solvent conditions. Further exploration of perforated layer would require more fine models that maps properties or real polymers including different rigidity, chemistry, influence of the surface, just to mention a few key parameters. Nevertheless, we believe that complex phase behavior, including stable PL phase, ripple structure and aggregates, discussed in this work by simple mesoscale model can provide a rational tool for targeting the proper surface modification and structure. Results and methods illustrated here also opens the way for exploring more complex systems such as polydisperse Y-shaped brushes in selective or bad solvent or under non-equilibrium conditions.

## Figures and Tables

**Figure 1 polymers-16-00721-f001:**
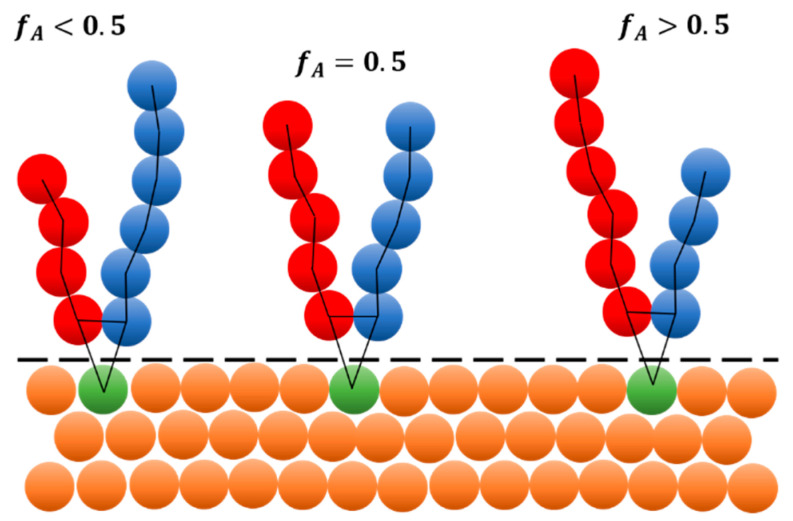
Mesoscale model of A/B Y-shaped brushes. Left: asymmetric brush with dominant B type, Middle: symmetric brush, Right: asymmetric brush with dominant A type. A/B homopolymer beads are represented by red (A) and blue (B) color, orange beads represent wall beads and green beads represent wall beads with grafted chains. Dashed line is a reflexive layer that avoids penetration of surrounding fluid (not displayed here) into the solid surface. Solid black lines that connect adjacent beads in the chain and polymer to the surface represent bonds. $f_{A}$ refers to ratio between length of homopolymer A and total brush length.

**Figure 2 polymers-16-00721-f002:**
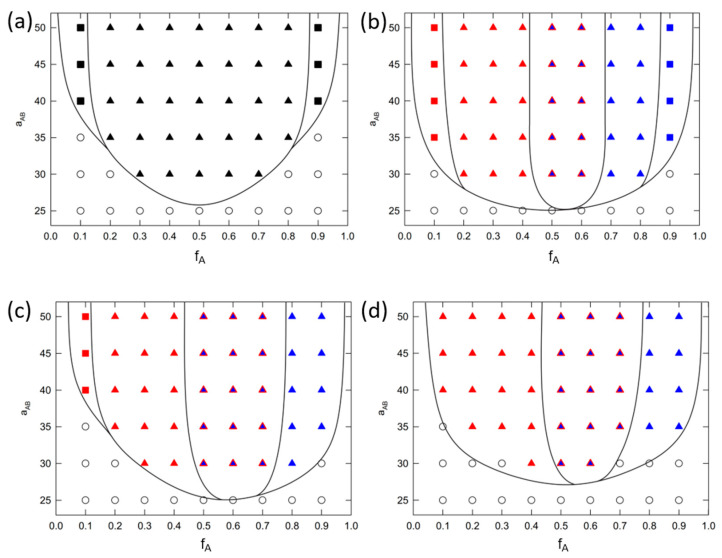
Phase diagrams of (**a**) monodisperse and partially polydisperse Y-shaped brushes where ${PDI}_{A}=1.0$  and (**b**) ${PDI}_{B}=1.1$, (**c**) ${PDI}_{B}=1.5$ and (**d**) ${PDI}_{B}=2.0$. Solid black lines indicate approximative phase boundaries and filled square stands for aggregates, triangles for ripple structure and triangles with two colors indicates perforated layer. Red color represents branch A assembly while blue color represents the opposite B branch. Black symbols represent monodisperse structures regardless of the brush type. Parameters $a_{AB}$ represent A-B incompatibility and $f_{A}$ represent ratio between homopolymer A chain length and total brush chain length. Phase diagrams in (**a**,**b**) were Adopted with permission from [20].

**Figure 3 polymers-16-00721-f003:**
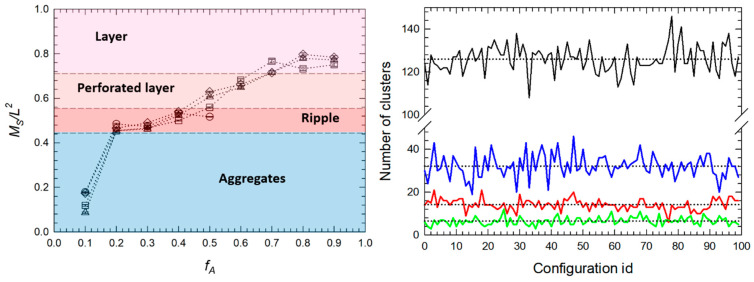
(**Left**): Minkowski characteristic $M_{S}$  as a function of Y-brush composition $f_{A}$. Results are shown only for A homopolymer. Dotted line serves as a guide to the eye. Circles represent monodisperse system, remaining symbols represent systems with polydisperse B homopolymer. Squares indicates low polydispersity ${PDI}_{B}=1.1$, triangles ${PDI}_{B}=1.5$ and diamonds reflect highly polydisperse system with ${PDI}_{B}=2.0$. Dashed line marks the approximative boundaries of different phases labeled by different color. The values of $M_{S}$ are scaled to the area of the surface, where $L=L_{x}=L_{y}=60r_{c}$. (**Right**): Evolution of number of clusters determined by DBSCAN clustering as a function of configuration id within the part of the trajectory. Profiles for system from Figure 2c (${PDI}_{A}=1.0$ and ${PDI}_{B}=1.5$) are shown with aggregates ($f_{A}=0.1$, black solid line), ripple structure ($f_{A}=0.3$, blue solid line), perforated layer ($f_{A}=0.6$, red solid line) and compact layer ($f_{A}=0.9$, green solid line), as a representative examples. Snapshots of clusters are shown in Appendix A. The dotted line represents an average number of clusters in the trajectory.

**Figure 4 polymers-16-00721-f004:**
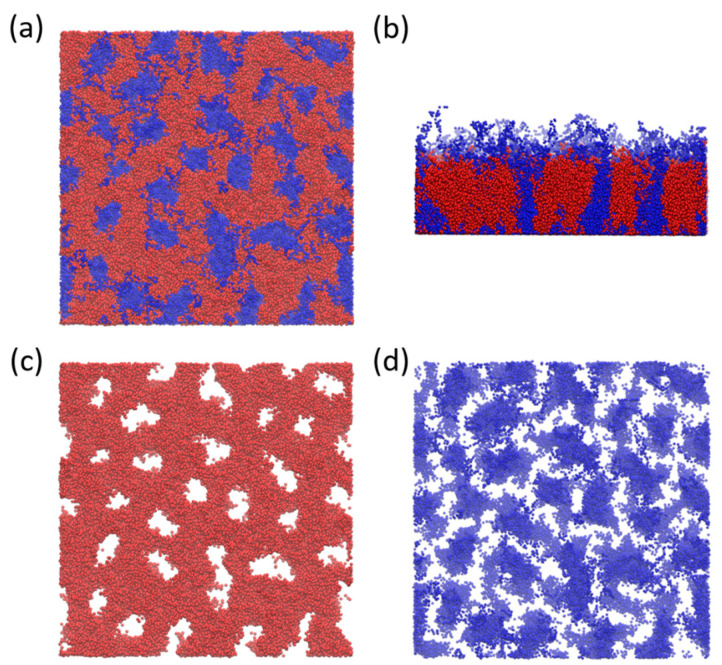
Simulation snapshot of perforated layer (PL) at $f_{A}=0.6$ for system from Figure 2c. (**a**) Top view of PL phase with both, A and B homopolymers. (**b**) Front view with B homopolymers penetrating the A compact layer. (**c**,**d**) shows top view of A homopolymer (red) and B homopolymer (blue) separately.

**Figure 5 polymers-16-00721-f005:**
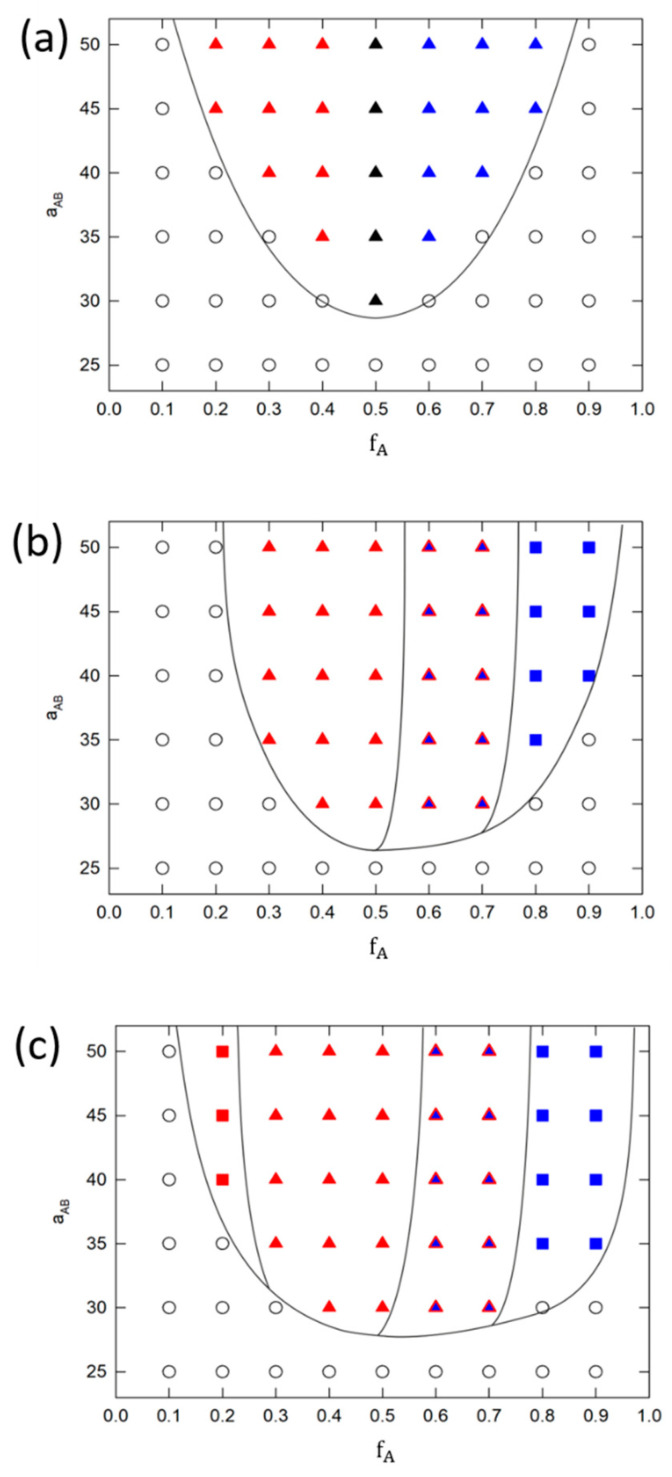
Fully polydisperse Y-shaped brushes phase diagrams where the polydispersity of A homopolymer is kept constant at 1.10 value and polydispersity of B homopolymer changes from (**a**) ${PDI}_{B}=1.10$, (**b**) ${PDI}_{B}=1.5$ to (**c**) ${PDI}_{B}=2.00$. Solid black lines indicate approximative phase boundaries and filled square stands for aggregates, triangles for ripple structure and triangles with two colors indicates perforated layer. Red color represents A homopolymer assemblies while blue color represents the B ones. Black symbols represent structures regardless of the homopolymer type Parameter *a**_AB_* represent A-B incompatibility and $f_{A}$ refers to ratio between homopolymer A chain length and total brush chain length.

**Table 1 polymers-16-00721-t001:** List of all systems considered in this study. ${PDI}_{A/B}$  represent values for individual homopolymers. Last column represents labels used for these systems further in the text. System with Asterisk were modelled by us previously in [20].

${PDI}_{A}$	${PDI}_{B}$	Label
1.00	1.00	S1 *
1.00	1.10	S2 *
1.00	1.50	S3
1.00	2.00	S4
1.10	1.10	S5
1.10	1.50	S6
1.10	2.00	S7
1.50	1.50	S8
1.50	2.00	S9
2.00	2.00	S10

## Data Availability

The raw data supporting the conclusions of this article will be made available by the authors on request.

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
