# Peer review of "Phase Behavior of Polydisperse Y-Shaped Polymer Brushes under Good Solvent Conditions"

_polymers, 2024, doi:10.3390/polym16050721_

Round 1

Reviewer 1 Report

Comments and Suggestions for Authors

see attached file

Reviewer 2 Report

Comments and Suggestions for Authors

This results in this manuscript will be of interest to scientists working on polymer brushes and are clearly explained. I am not aware of other publications that deal with polydispersity in this form of polymer brush as thoroughly as this study. The technique used is an established one, and the simulation procedure is reasonable and described in enough detail to allow reproduction of the results. The parameter values used are realistic and in line with other studies using this technique. There are some small errors in the English, and the gamma function is referred to as the delta function in line 211, but the overall standard of presentation is good. The figures are very clear, but the use of S1, S2, etc. to label both the samples and the figures in the ESI is slightly unhelpful at times and I would suggest changing the labelling of the samples. Overall, this is a careful and interesting study, and I am happy to recommend publication.

Comments on the Quality of English Language

The paper is clearly written. However, there are several small errors in spelling (e.g., 'chian' instead of 'chain' in line 214) and grammar (e.g., 'tethered to surface' instead of 'tethered to a surface' in line 25).

Reviewer 3 Report

Comments and Suggestions for Authors

For the simulation works, kindly provide supportive experimental works for comparison in order to validate the results. If not, please provide the limitations of study. You also need to provide a table of comparison what is the improvement of the simulation model that you have done and compared to the existing results. Then this can make your results to be more meaningful. 

In your abstract, please provide also the limitation of this study instead of just mentioned the positive part of your works. 

Please provide the list of nomenclature use so that the readers can easily to refers to it.

Round 2

Reviewer 1 Report

Comments and Suggestions for Authors

Reviewer 3 Report

Comments and Suggestions for Authors

The paper now is accepted for publication

Round 3

Reviewer 1 Report

Comments and Suggestions for Authors

I recommend the acceptance of the manuscript for publication.